# Towards a New System for the Assessment of the Quality in Care Pathways: An Overview of Systematic Reviews

**DOI:** 10.3390/ijerph17228634

**Published:** 2020-11-20

**Authors:** Roberto Latina, Katia Salomone, Daniela D’Angelo, Daniela Coclite, Greta Castellini, Silvia Gianola, Alice Fauci, Antonello Napoletano, Laura Iacorossi, Primiano Iannone

**Affiliations:** 1National Center for Clinical Excellence, Healthcare Quality and Safety, Istituto Superiore di Sanità, 00162 Rome, Italy; roberto.latina@uniroma1.it (R.L.); katia.salomone@guest.iss.it (K.S.); daniela.dangelo@iss.it (D.D.); daniela.coclite@iss.it (D.C.); alice.fauci@iss.it (A.F.); antonello.napoletano@iss.it (A.N.); primiano.iannone@iss.it (P.I.); 2Unit of Clinical Epidemiology, IRCCS Istituto Ortopedico Galeazzi, 20161 Milan, Italy; greta.castellini@grupposandonato.it (G.C.); silvia.gianola@grupposandonato.it (S.G.)

**Keywords:** clinical pathways, clinical decision support system, quality assessment, healthcare, clinical practice guidelines

## Abstract

Clinical or care pathways are developed by a multidisciplinary team of healthcare practitioners, based on clinical evidence, and standardized processes. The evaluation of their framework/content quality is unclear. The aim of this study was to describe which tools and domains are able to critically evaluate the quality of clinical/care pathways. An overview of systematic reviews was conducted, according to Preferred Reporting Items for Systematic Reviews and Meta-Analyses, using Medline, Embase, Science Citation Index, PsychInfo, CINAHL, and Cochrane Library, from 2015 to 2020, and with snowballing methods. The quality of the reviews was assessed with Assessment the Methodology of Systematic Review (AMSTAR-2) and categorized with The Leuven Clinical Pathway Compass for the definition of the five domains: processes, service, clinical, team, and financial. We found nine reviews. Three achieved a high level of quality with AMSTAR-2. The areas classified according to The Leuven Clinical Pathway Compass were: 9.7% team multidisciplinary involvement, 13.2% clinical (morbidity/mortality), 44.3% process (continuity-clinical integration, transitional), 5.6% financial (length of stay), and 27.0% service (patient-/family-centered care). Overall, none of the 300 instruments retrieved could be considered a gold standard mainly because they did not cover all the critical pathway domains outlined by Leuven and Health Technology Assessment. This overview shows important insights for the definition of a multiprinciple framework of core domains for assessing the quality of pathways. The core domains should consider general critical aspects common to all pathways, but it is necessary to define specific domains for specific diseases, fast pathways, and adapting the tool to the cultural and organizational characteristics of the health system of each country.

## 1. Introduction

The European Pathway Association defines clinical/care pathways (CPs) as a methodology for the mutual decision-making and organization of care processes for a specific cohort of patients during a defined period [1]. Clinical pathways and integrated care models are increasingly used in different European and worldwide healthcare settings, such as acute hospitals, rehabilitation centers, and primary care facilities [2,3,4,5]. Clinical pathways, also defined as integrated care pathways (ICP) [6], are being used in different health care systems, primarily to improve the efficiency of hospital care and the quality of care by reducing variations in the care processes and outcomes [7]. A CP is not defined as a simple tool, but it is a complex intervention that is more than just a piece of paper or document, consisting of multiple components working interactively [8,9]. The core characteristics of pathways include many aspects: an explicit statement of goals, a multidisciplinary team of clinicians and managers, facilitation of communication and coordination of roles, the need to be built on evidence-based best practice, and patient/relative involvement and expectations. CP involves operational research and methodologies, with the use of documentation, dataset, and transparent and standardized quality assessment [10,11]; collaboration with the general practitioner and follow-up; and its impact on the organization of clinical processes and outcomes [12,13]. Nevertheless, the effectiveness of CPs cannot be generalized because of the insufficient number of controlled studies and because of confounding factors and sources of contamination affecting the validity of the outcomes [5]. CPs do not always guarantee that a care process will be perfectly organized, and a lack of consensus regarding research outcomes is not surprising given the lack of agreement regarding structure, indicators, or domains that define a CP [14]. One of the important traps in path development is the lack of evidence-based key interventions and outcome indicators [15], and measuring integrated care is difficult due to the lack of suitable tools to measure the different aspects of integration, e.g., transition from hospital to primary care [16]. There is a meaningful interest in CPs among health care practitioners, but they are among the least used clinical decision support instruments [17]. The biggest problem of innovation in care practices is represented by the translation of theoretical “good practices” into operators’ daily practices; in other words, the transition from “theoretical” efficacy to “practical” effectiveness in the real world, outside the experimental context. However, it is controversial whether the implementation of CPs is an economic process [18] or not [19], and CP appraisal tools describe the structure of the path and the organization of the care methodology, but they infrequently evaluate the content of the CP, quality, and outcomes. Another important domain includes the economic aspects, as rationalization and optimal resource use, with particular concern for the health professionals, is one of the main strategic goals for which CPs are designed and implemented. However, as the well-known maxim states, if you cannot measure it, you cannot manage it [20]. When CPs are used, the rigorous evaluation of the core framework and its implementation is a bare necessity, which includes the ability to be repeated even when the process is over, in particular if it is necessary to assess the health outcomes. Therefore, attention should be paid to the definition of the results to be achieved through the identification of specific indicators so that the results are truly measurable and suitable for the pursuit of the objectives set. The monitoring of quality, structure, process, outcome [21], and safety indicators makes it possible to identify critical areas for improvement. Overall, an accurate and robust framework for the evaluation of the CPs can be compliant with the health technology assessment (HTA) approach, as it is based on multidisciplinarity and it considers a multi-domain evaluation. Of course, a set of specific criteria and indicators is requested in order to evaluate the clinical pathways along the traditional HTA domains: effectiveness, safety, organizational, economic, ethical, legal, and social aspects [10]. Since several tools have been adopted to assess the quality of CPs, we decided to conduct an overview of systematic reviews to combine, appraise, and summarize the results of related systematic reviews.

## 2. Methods

### 2.1. Research Question

Which are the essential tools and domains used to assess the quality of CPs in terms of structure, process, and outcome?

### 2.2. Search Strategy

An overview of a systematic review (SR) of the literature was conducted, in agreement with the Preferred Reporting Items for Systematic Reviews and Meta-Analyses (PRISMA) method (Figure 1). The records were identified using an advanced search strategy on different electronic databases, such as Medline, Embase, Science Citation Index, PsychInfo, CINAHL, and Cochrane Library, through keywords, terms, and medical subject headings (Mesh) combined with each other through the Boolean operators “AND” and “OR”(Figure 2).

### 2.3. Definition of Measurement Instruments

We defined any measurement devices or measurement tools, such as questionnaires, checklists, or observation forms, that can be filled out by researchers, clinicians, administrators, or patient representatives to assess the quality of structures, processes, or outcomes associated with domain(s) or criteria/indicator(s), such as patient engagement.

### 2.4. Eligibility Criteria

Reviews that observed the main objective of describing the CP quality assessment tools or indicators or domains were included. Systematic reviews were evaluated, having to cover: (1) tools for evaluating CPs; and (2) the attributes and/or criteria and/or domains that measure the results and quality of a CP, and their ability to evaluate the quality of a CP critically. It was decided not to restrict the search field, in order to collect as much information as possible. Restrictions related to the language (English/Italian), the type of study (systematic reviews, meta-analyses), and publication date in the last 5 years (from the beginning of 2015 to July 2020) were used. Snowballing was undertaken, starting from the included reviews and from the references of each article. This method allowed us to find more specific literature about a subject rapidly and relatively effortlessly.

### 2.5. Data Extraction

A total of 3464 records from these systematic reviews were identified (Figure 1). Duplicates were eliminated, using a reference manager. The publications were initially subjected to a screening process conducted by two independent researchers, aimed at assessing their potential compliance with the eligibility criteria. The titles were evaluated first and then the abstracts. In the event of a discrepancy, the results were discussed until an acceptable degree of concordance was reached. At the end of the screening of the abstracts, the full texts of the relevant publications were found and analyzed, and included if appropriate. Furthermore, each indicator was classified using the domains/areas according to The Leuven Clinical Pathway Compass (TLCPC-2003) [22]. The TLCPC is constituted by five areas to subdivide criteria: clinical (e.g., mortality rate), financial (e.g., length of stay), process (e.g., evidence-based practice and guidelines), service (e.g., patient centered care), and team indicators (e.g., multidisciplinary involvement). The TLCPC is a conceptual framework developed to assess the effect of CPs and to follow-up a patient population. Indicators were then classified according to the Donabedian model (structure, process, and outcome) [23] by two independent researchers, until complete agreement was established.

### 2.6. Quality Appraisal of Reviews

For quality appraisal of each review, the Assessment the Methodology of Systematic Review (AMSTAR-2) methodological quality of systematic reviews measure tool was used [24]. This instrument is composed by 16 items, 7 of which are critical, and including protocol, assessment of risk of bias, justification of including/excluding studies, appropriate meta-analytic methods, and consideration of impact of publication bias. Four options can be used to answer each item questions: “yes”, “partial yes”, “no”, or “no meta-analysis conducted”. Moreover, based on critical items, the quality of systematic reviews can be classified into different levels: “High”: zero or one non-critical weakness. The systematic review provides an accurate and comprehensive summary of the results of the available studies that score and address the question of interest; “moderate”: more than one non-critical weakness. The systematic review has more than one weakness, but no critical flaws. It may provide an accurate summary of the results of the available studies that were included in the review; “low”: One critical flaw with or without non-critical weaknesses. The review has a critical flaw and may not provide an accurate and comprehensive summary of the available studies that address the question of interest; and “critically low”: more than one critical flaw with or without non-critical weaknesses. The review has more than one critical flaw and should not be relied on to provide an accurate and comprehensive summary of the available study. Multiple non-critical weaknesses may diminish confidence in the review, and it may be appropriate to move the overall appraisal down from moderate to low confidence.

We recorded the protocol for this review in the PROSPERO, which was approved, and it is publicly available under the registration number CRD42020210486.

### 2.7. Statistical Analysis

Data analysis included descriptive measures (e.g., counts and frequencies) of the domains and indicators used to evaluate the clinical pathway/integrated care.

## 3. Results

From the 3464 articles identified, 1544 duplicates were eliminated and, out of the remaining 1920, 4 relevant reviews were obtained, while 5 reviews were obtained through snowballing (Figure 1). Of these, six reviews relate to integrated care (IC), three reviews relate to clinical pathways (CPs), and one study was explicitly addressed to the pediatric population (Table 1).

### 3.1. Terminology and Definitions of CP

According to the results of this review, conducting studies on methods for measurement of PC/ICP is a difficult charge, because the conceptual diversity used within the fields and definitions of both is vast. While studies aimed at CPs are able to explicitly describe the constituent domains and indicators, studies relating to IC focus on indicators of continuity and integration of care and care processes (Table 2).

### 3.2. Synthesis of Reviews Included

The oldest review (Vanhaecht et al., 2006) [25] describes 15 tools applied specifically to CP, of which 7 tools were published or developed in or after 1998, and describes only 7 tools in depth. The variability of characteristics confirms a scarcity of consensus on the concept and definition of CPs. Only one tool, the Integrated Care Pathway appraisal tool (ICPAT) of Whittle et al., (2004) [13], was validated. The two systematic reviews on CPs for gastrointestinal surgery (Lemmens et al., 2008) [26] and colorectal cancer surgery (van Zelm et al., 2018) [27] do not describe tools but indicators. The criteria that were used to evaluate clinical pathways in the literature were scored for five domains of TLCPC [22]. The most common one is “financial”, which assesses the ability of CP to reduce the length of stay truly adverse effects on clinical outcome. The next most common is “clinical”, as complications and re-admission rates are commonly measured. Many clinical criteria are specific and applicable in different clinical conditions or related to surgery (admission to intensive care unit (ICU), rates of mortality). However, the authors underline that major attention is needed to indicators of patient safety [27], using Enhanced Recovery After Surgery (ERAS) as a fast multimodal pathway, suggested that the different components are applicable.

The review by Strandberg-Larsen et al., (2009) [28] identified 24 papers, published from 1979 to 2007. The authors concluded that the methods surely reflected the conceptual diversity used within the area. There was no consensus on which information sources best captured integrated healthcare delivery. The degree of internal validity was described in 9 of 19 articles published and none had been thoroughly validated across different settings.

The review by Lyngsø et al., (2014) [29] found 23 measurement tools that aimed to achieve the pivotal role of assessing the impact of integrated care models focusing on the degree of integration based on core organizational elements. All studies include defined structural and process aspects and six included cultural aspects; additionally, they found different components for constructing health–system integration independent of the healthcare system or population care: patient focus, information technology, leadership and organizational culture, commitments and incentives to deliver integrated care, clinicals (protocols and clinical guidelines), quality improvement/performance measurement, and financial incentives.

The review of Uijen et al., (2012) [30] included 21 questionnaires measuring continuity of care. Seventeen instruments assessed continuity of care from the viewpoint of the patient, and four from the position of the care practitioner and program manager. From the tools measuring these aspects, three were created for mental illness, cancer, for complex and chronic care needs, heart failure, and others. Other (10) tools assessed aspects of staff, health practitioners, and cross-boundary continuity. Most questionnaires were originally developed in English, eight of which were exclusively appropriated for the primary care setting and eight for the secondary care setting; five were developed for both settings. The methodological quality of the tools concerning these psychometric characteristics in terms of reliability was generally poor, and validity (cultural and criterion) was not evaluated in each of the studies.

The systematic review by Bautista et al., (2016) [31] describes 209 instruments involving work from Europe and the USA/Canada, Africa, and Asia-Pacific. They are available to assess integrated care, with different psychometric characteristics, in terms of reliability, validity, and responsiveness, which were validated in the primary care population. In addition, many tools implicated subjects affected by non-chronic conditions, multiple chronic conditions, tumors, and mental disorders. The majority of the instruments measured different domains, such as care integration and patient-centered care. They as well codified, and the tools show agreement with the different domains of the six dimensions of the Rainbow Model of Integrated Care. The results emphasize the need for research of more high-quality studies that may measure the instrument’s features and quality. This systematic review of the measurement properties of instruments does not directly answer how services can be integrated or how patients want care to be delivered. It also does not provide evidence of the effectiveness of integrated care (outcomes).

The SR by Suter et al., (2017) [32] obtained 16 indicators/domains (Table 2) included in instruments that measure health system integration. In total, 94 of the tools cover observational tools, questionnaires, and other types of instruments (checklists, and indicators). Ninety-two were based on self-report, 56 completed by providers, 42 by inpatients, 10 by administrators, and 6 by either or all. Many tools were created and validated with specific patients (e.g., pediatric, others) but could conceivably be borrowed for the general population. Seventeen instruments measured continuity of transitional care covering the care continuum. Some domains aimed to catch the quality and transition planning as experienced by the life experience of the patient between acute and primary care, and other community care services and settings. Some instruments assessed the structure, process, and outcome areas mostly from the patient or family point of view (i.e., administrative processes, customer satisfaction, communication, level of empowerment and empathy). This review highlights that care pathways should develop consistency and continuity of care across staff members. Moreover, it is essential to assess performance management, how systems share information across sectors, and whether electronic clinical records are shared across different settings. Other assessments are on organizational culture and leadership, physician integration within care teams and across sectors, and financial management, with the aim of capturing whether there is connection among organizational aims and how resources are being used. Resource allocation best practice questions cover priority-setting methods, such as grants, funding, service monitoring, and outcome evaluation.

The article of Valentijn et al., (2019) [33] describes the development of the Rainbow Model of Integrated Care (RMIC-MT), both patient and provider versions, which were based on the patient-reported outcome measures. The RMIC-MTs are helpful tools for evaluating the care coordination process on integrated renal care. Each tool may be administered to care practitioners and patients, respectively. The RMIC-MTs identify four core domains (person-centeredness, and service, professional, and organizational coordination), four auxiliary (community-centeredness, technical and cultural competence, system context), and integrated care domains. The RMIC-MTs may be used as a pre/post intervention to measure the impact of an integrated care program, as an ongoing performance assessment tool to focus on core points of integrated service delivery across staff and organizations.

### 3.3. Prevalent Domains and Categorization of Indicators with Donabedian’s Model

Table 2 describes the essential characteristics of domains and/or those indicators that are evaluated by the instruments summarized. According to the Donabedian classification, 20.2% (*n* = 15) were structural indicators, 50.0% (*n* = 37) process indicators, and 29.7% (*n* = 22) outcome indicators.

### 3.4. Instruments/Tools

This review revealed a number of useful tools for evaluating the quality of CPs or ICs in well over 300 cases. It is not possible to aggregate the results and describe them individually, due to their heterogeneity with respect to the contents, domains, and indicators they use. The results relating to the areas and domains using The Leuven Clinical Pathway Compass are summarized in Figure 3 and Table 2. The reviews of Valentijn et al., (2019) [33] and Lyngsø et al., (2014) [29] were excluded from these results because it is not possible to calculate the frequency of each domain of each instrument. Researchers should refer to the original documents for greater clarity.

### 3.5. Quality Appraisal of Reviews

The results relating to critical quality appraisal using AMSTAR-2 are tabulated in Table 3. Only three reviews are of high quality according to AMSTAR-2. The others are moderate and low quality. 

## 4. Discussion

The purpose of this overview of systematic reviews was to describe the domains, contents, and criteria/indicators, through a tool applicable in the evaluation of the quality of a clinical pathway in order to identify a set of indicators that could be integrated in an evaluation framework, consistently with the HTA approach. Standardizing care through the use of CP is an important method to diminish the heterogeneity of the clinical process, reduce malpractice risk, guarantee high standard of care and high quality, and reduce costs [34]. According to the results of this review, definitions of CP and IC seem to not always be the same thing, since the conceptual diversity used within the field and definition of both is huge, although care or critical pathways are described as integrated care pathways [6]. The confusing and interchangeable terminology and frequently ambiguous use of terms is a challenge for the systematic approach needed when conducting SR. In fact, in the study of de Luc et al., (2001) [35], 17 different terms were used for clinical pathways, as described in another study [36]. This variation in the terms used, so referring to a complex intervention without standardizing its elements, only adds confusion towards the achievement of a standard definition [37]. The often-discussed conceptual ambiguity of CP/IC is a crucial key in assessing the areas of integration. Generally, “clinical pathway” is used for the pathway within a 24-h department of a hospital. A “care pathway” is longer and includes outpatient care, after discharge from the hospital [38]. CPs are part of IC, and they are a method to fulfill an outcome, and a complex intervention to guarantee the structure, the interdisciplinary staff process, and the follow-up of the outcomes of a specific clinical care process. “Integrated care” is a coherent set of methods and models on the funding, administrative, organizational, service delivery, and clinical levels designed to create connectivity, alignment, and collaboration within and between the cure and care sectors” [32,39]. Nevertheless, a lot of instruments were obtained from this review (over 300, data not shown) as questionnaires, tools contained checklists and indicators, and the first audit instruments were implemented from the 1990s [40].To date, there is no gold standard tool to assess the quality of CP/IC or a fast CP [27]. It suggests important gaps in CP/IC quality tool measurement that underscore the lack of a standard measurement instrument and the integration of care across healthcare systems. The term gold standard refers to an available benchmark as it is not the perfect test but solely the best available and reliable one that has a standard with known results [41]. In our case, it is necessary to evaluate a complex process, which takes into account the different approaches available in the literature, integrated with each other, as described in Figure 4. This multi-principle framework proposal (Figure 4) arises from the Donabedian model [23] integrated with that of HTA [42], The Leuven Clinical Pathway Compass [22], and RMIC-MT [33], adapted to the health service model in each country. This proposal could guide the construction of a multi-criteria model to define a gold standard and related indicators for assessment of CP/ICP, in its completeness and complexity. In this direction, the Medical Research Council has made guidance on process evaluation of complex interventions [43]. This guidance delineates and links the key pivotal components of process assessment of complex interventions. Process evaluations cannot provide answers to all of the uncertainties of a complex intervention [44]. It is likely that in a CP/IC, it is necessary to identify: (1) standardizable core domains/indicators in each pathway (homogenous patients group, diagnosis related group), (2) standardizable core domains/indicators independently of the clinical condition, and (3) standardizable core domains/indicators in a fast multimodal pathway, e.g., the Enhanced Recovery After Surgery (ERAS) [27,45]. Moreover, a CP relating to the emergency area can be recommendable, in relation to outpatient CPs, as an encouraging approach to reduce hospital admission after emergency care [34]. De Bleser et al. [36] proposed interesting core characteristics, aims, and outcomes for analysis of the definition of CP. It is also probable that each health system chooses to focus on specific aspects/domains of CP/IC while dispensing with other aspects, given their respective priorities and prevailing needs, goals, and standards in each healthcare delivery. In Italy, for example, in the National Health System, the evaluation of CPs at regional and national level has to be done in terms of appropriateness, clinical outcome, equity, and economic impact [46] for each different disease, and it seems to ignore the integration of care. Moreover, systematic measurement methods are vital in continuously developing the knowledge base of integrated care.

This overview of systematic reviews shows some critical issues. Firstly, only three out of nine of these reviews, according to AMSTAR-2, are of “high” quality (Table 3), while the other reviews have moderate and low or critically low quality. It is possible that the methodological rigor is more evident in the reviews published from 2016 onwards, or that the methodological rigor in place was not explicitly described. A systematic review needs to use a transparently formulated query that uses systematic and rigorous methods to recognize, collect and select, and critically appraise relevant research, and to analyze information from each of the studies that are enclosed [47]. It is used to explore the best evidence of clinical safety or effectiveness of healthcare interventions. Secondly, deficient aspects related to stakeholder involvement are described, and low adherence criteria are described by tools related to evidence-based models. A CP specifically states the outcomes and pivotal elements of care based on evidence-based practice guidelines, and patient desires [36]. The development and validation of evidence-based critical pathways for inpatient and outpatient settings is a research and clinical priority. A transparent system of medical justice that acknowledges clinical guidelines when determining standards of care could help reduce health practitioners’ perceived risk [34,48]. Thirdly, although most questionnaires are culturally effective, psychometric validity and responsiveness were not evaluated or they are of low to moderate quality in another study [30,31]. Moreover, many instruments with desirable psychometric properties can have too many items to be practical, so they can be used for research but are not used in routine clinical practice. Many tools are individually capable of measurement and some domains are specific to a particular CP. Fourthly, the management of follow-up is unclear. A transmural pathway is even longer than others, and covers the preliminary and the follow-up process in primary care or another care facility [49]. Finally, although it is difficult to compare the reviews, they share the need to orient the care toward patient-centeredness. A recent review describes how several initiatives that have implemented patient participation point to its positive effects on quality of care, increased patient satisfaction and safety, and healthcare providers appearing to have more empathy, which in turn results in better informed and empowered patients [50]. About the results of the classification of the indicators according to Donabedian, it emerged that the process and outcome indicators are more sensitive to the clinical conditions or settings (medical or surgical) than the structural indicators, which can be more easily standardized. Moreover, HTA represents an accurate methodological framework aimed at rationalizing the allocation of a care pathway, through the provision of information regarding the effectiveness, safety, and use of health technologies, as well as ethical, societal, and organizational domains [42].

The present study has some methodological limitations. Firstly, we only reviewed systematically literature published within a short 5-year time period. This limitation was partially addressed through the use of snowballing. Secondly, we included only two languages.

## 5. Conclusions

This overview further confirms the need for a more standard process of selecting appropriate tools to assess different core constructs/domains for CP and IC, whereas the boundary between the two pathways is not always clearly defined. It is clear that involvement of patients/family is an essential aspect, together with multidisciplinarity and interdisciplinarity. The proposal of a multi-criteria framework could become a guide to define a gold standard and related indicators for the assessment of CP/ICP. Nevertheless, it is imperative to develop valid measures of assessment for the content of the path, integration across healthcare systems, and clinical and economic outcomes. Overall, it is worthwhile to keep in mind that each CP/IC refers to specific clinical conditions or disease cohorts. Therefore, the process indicators are specific and are closely related to the guidelines. Determining at least two gold standards could be useful, one general and one for a specific disease, to standardize and evaluate the ever more complex treatment processes while also considering cost-effectiveness aspects.

## Figures and Tables

**Figure 1 ijerph-17-08634-f001:**
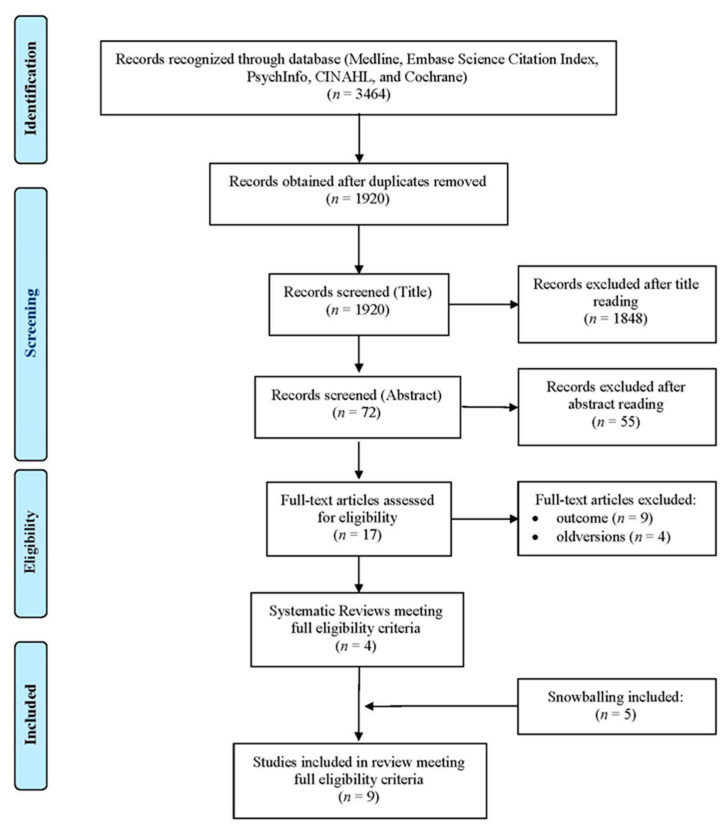
Flow diagram.

**Figure 2 ijerph-17-08634-f002:**
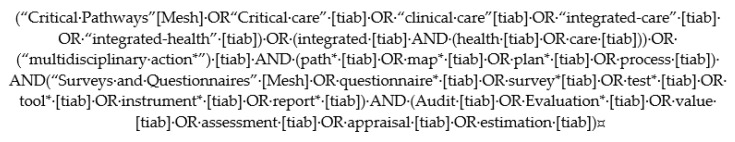
Query used on Pubmed.

**Figure 3 ijerph-17-08634-f003:**
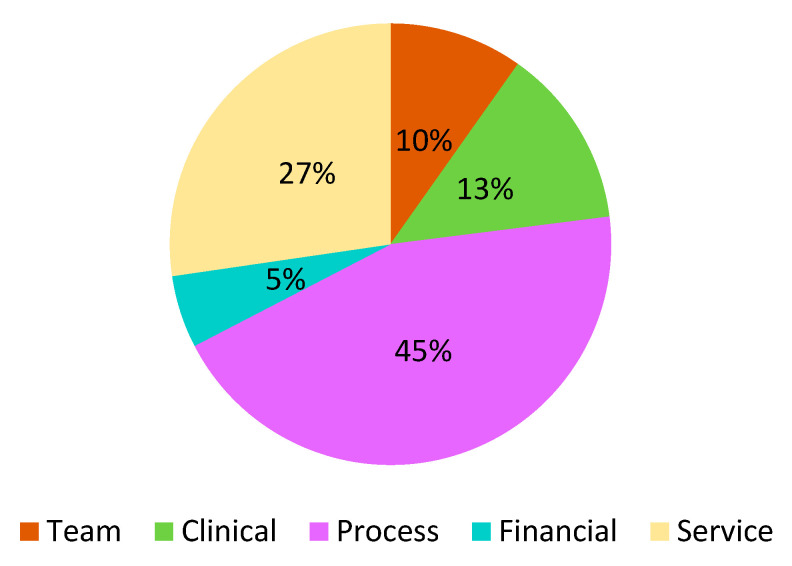
Domains using The Leuven Clinical Pathway Compass.

**Figure 4 ijerph-17-08634-f004:**
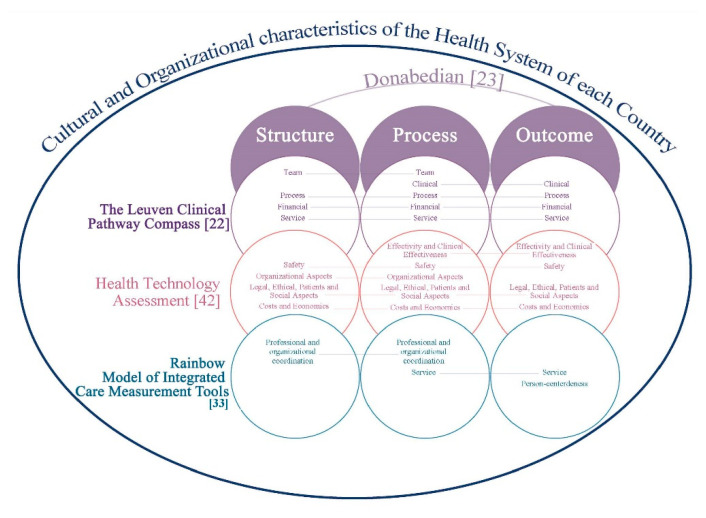
Multicriteria approach to assess the quality of care or integrative care pathways.

**Table 1 ijerph-17-08634-t001:** Reviewsincluded.

n.	Authors (Year)	Title	Focus
1	Vanhaecht et al., (2006) [25]	Clinical pathway audit tools: a systematic review	Clinical Pathways
2	Lemmens et al., (2008) [26]	Systematic review: indicators to evaluate effectiveness of clinical pathways for gastrointestinal surgery	Clinical Pathways
3	van Zelm et al., (2018) [27]	Development of a model care pathway for adults undergoing colorectal cancer surgery: evidence-based key interventions and indicators	Clinical Pathways
4	Strandberg-Larsen et al., (2009) [28]	Measurement of integrated healthcare delivery: a systematic review of methods and future research directions	Integrated Care
5	Lyngsø et al., (2014) [29]	Instruments to assess integrated care: a systematic review	Integrated Care
6	Uijen et al., (2012) [30]	Measurement properties of questionnaires measuring continuity of care: a systematic review	Integrated Health Care
7	Bautista et al., (2016) [31]	Instruments measuring integrated care: a systematic review of measurement properties	Integrated Care
8	Suter et al., (2017) [32]	Indicators and measurement tools for health systems integration: a knowledge synthesis	Integrated Care
9	Valentijn et al., (2019) [33]	Validation of the Rainbow Model of Integrated Care Measurement Tools (RMIC-MTs) in renal care for patient and care providers	Integrated Care

**Table 2 ijerph-17-08634-t002:** Indicators used to evaluate the clinical pathway/integrated care in each domain.

The Leuven Clinical Pathway Compass (2003) [22]	1. Vanhaecht et al., (2006) [25]	2. Lemmens et al., (2008) [26]	3. van Zelm et al., (2016) [27]	4. Strandberg-Larsen et al., (2009) [28]	5. Uijen et al., (2012) [30]	6. Bautista et al., (2016) [31]	7. Suter et al., (2017) [32]	Total
**DOMAINS and Indicators**	*n*	*n*	*n*	*n*	*n*	*n*	*n*	*n* (%)
**(1) Team**								0
Influence on team satisfaction		1						1
Multidisciplinary involvement/team effectiveness	7				10		12	29
Physician integration within care teams/professional Integration/nursing care integration				5		11	6	22
Supporting services (i.e., education and social services)			14	6				20
Total Team								72 (9.8)
**(2) Clinical**								
Complication rate			14					14
Compliance to protocol			1					1
Discharge destination		15						15
Mortality		10	6					16
Number of admissions or length of stay on HDU/ICU *		4						4
Number of complications/post-operative morbidity		16						16
Number of re-admissions		3						3
Number of re-operations		2						2
Pain scores/post-operative pain		2	3					5
Post-operative treatment with fluids		2						2
Readmission rate (<30 days)								0
Removal of bladder catheter		2	3					5
Reoperation/intervention		2						2
Time taken to mobilize			11					11
Use of intravenous catheter			1					1
Stress Index			1					1
Total Clinical								98 (13.2)
**(3) Process**								0
Accountability	4							4
Appropriate use of antibiotics/others		1						1
Completeness and quality of documentation/data tracked and shared with stakeholders/informatic integration	3	1		3			1	8
Clinical outcomes being measured							2	2
Care continuity/clinical integration				8	17	172		197
EBM/guideline	5							5
Implementation of pathway	3							3
Maintenance of pathway	5							5
Number of clinical examinations (labs, radiology)		3						3
Organizational goals and objectives aligned across sectors							1	1
Outcome management	7							7
Performance measurement domains and tools in place							2	2
Primary care network structures					8		8	16
Safety (risk management)	7							7
Transitional (transferring care from one area to another)							17	17
Use of shared clinical pathways across the continuum of healthcare (e.g., diabetes, asthma care) Organizational integration or coordination and specialty/or combination				4		40	7	51
Total Process								329 (44.4)
**(4) Financial**								
Attainment of goals and objectives are supported by funding and human resource allocation							1	1
Influence on length of stay		20	15					35
Influence on length of stay HDU/ICU			1					1
Influence on medical costs		5						5
Total Financial								42 (5.3)
**(5) Service**								
Influence on patient satisfaction		6	1					7
Individualization of care pathways for patients with co-morbidities							7	7
Patient centered care/family involvement in care planning	3					148	34	185
Quality of life (SF-36)		1	1					2
Total Service								201 (27.3)

* HDU: High Dependency Unit; ICU: Intensive Care Unit.

**Table 3 ijerph-17-08634-t003:** Qualityappraisal.

n.	Authors	Items of AMSTAR 2																Overall Rating Quality
		1	2 *	3	4 *	5	6	7	8	9 *	10	11 *	12	13 *	14 **	15 *	16	
1	Vanhaecht et al., (2006) [25]	NA	PY	Y	PY	Y	Y	Y	Y	NA	N	NA	NA	NA	Y	Y	N	Low
2	Lemmens et al., (2008) [26]	NA	PY	Y	PY	Y	Y	Y	Y	NA	Y	NA	NA	NA	Y	Y	PY	Moderate
3	van Zelm et al., (2018) [27]	NA	PY	Y	Y	Y	Y	Y	Y	NA	Y	NA	NA	NA	Y	Y	N	High
4	Strandberg-Larsen et al., (2009) [28]	NA	PY	Y	Y	Y	Y	Y	Y	NA	Y	NA	NA	NA	Y	N	N	Low
5	Lyngsø et al., (2014) [29]	NA	PY	Y	PY	N	N	PY	PY	NA	N	NA	NA	NA	Y	Y	N	Critically Low
6	Uijen et al., (2012) [30]	NA	PY	Y	PY	Y	Y	Y	Y	NA	Y	NA	NA	NA	Y	Y	Y	Moderate
7	Bautista et al., (2016) [31]	NA	PY	Y	Y	Y	Y	Y	Y	NA	Y	NA	NA	NA	Y	Y	Y	High
8	Suter et al., (2017) [32]	NA	PY	Y	PY	Y	Y	Y	Y	NA	Y	NA	NA	NA	Y	Y	Y	Moderate
9	Valentijn et al., (2019) [33]	NA	PY	Y	Y	Y	Y	PY	Y	NA	Y	NA	NA	NA	Y	Y	Y	High

Note: * Critical items ** Heterogeneity related by comparing the results of each of the studies included. NA: not applicable. PY: Partial Yes. Y: Yes. N: Not.

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
