# Peer review of "Towards a New System for the Assessment of the Quality in Care Pathways: An Overview of Systematic Reviews"

_ijerph, 2020, doi:10.3390/ijerph17228634_

Round 1

Reviewer 1 Report

This is a systematic review of factors related to the quality of clinical pathways where the authors propose a new direction in quality assessments. The following comments and recommendations are forwarded in order to strengthen a somewhat disjointed strategy for a new and improved assessment paradigm. However, the authors are correct in stating that the current state of the art of quality assessments of clinical pathways or guidelines is in need to a more standardized approach.

Title: clearly describes the nature of the study.

Abstract: line 23, by what standard did you conclude that no "gold standard" exists for assessment instruments? What are elements of a gold standard in this context?

Key words: integrated care pathway is not a MeSH term. Quality, assessment and review are too generic to be useful as search terms. Consider replacing integrated care pathways with 'clinical decision support systems', and replacing quality, assessment, and review with 'quality assessment, healthcare' and 'clinical practice guideline.'

Line 82 needs a citation for Health Technology Assessment.

Figure 3 would be more readable if presented in color rather than black and white.

Still, it is unclear to me why the authors think that there is no 'gold standard' for the assessment of clinical pathways or guidelines since the Donabedian model has been utilized for healthcare assessment since first proposed in 1966. Since it only expanded the specificity of structure, process, and outcome variables, why was Leuven or any other paradigms needed? If the authors are correct that standard definitions used in evaluating the quality of organized care interventions doesn't exist, perhaps a more organized proposal to operationally define key measurement terms, such as team integration or patient engagement, would provide a the reader with elements of a more robust conceptual framework. Since they are suggesting an integration of the Rainbow Model of Care, the Leuven Clinical Pathway Compass, and the Health Technology Assessment methodologies and definitions, perhaps a chart or figure to illustrate this new paradigm configuration would be helpful to the reader. Consider focusing the discussion on how you would integrate these three frameworks to meet the Medical Research Council's guidance on process evaluation of complex interventions. In general, the discussion needs more focus and specificity based on what you are proposing based on your analysis of incongruities and inconsistencies in current literature.

Reference are in mdpi style.

Thank you for the opportunity to read and comment on your important work.

Author Response

Please, see the table

Reviewer 2 Report

This paper presents an overview of systematic reviews on tools to evaluate the quality of care pathways.

Suggestions and questions (answers can/should be used to improve the paper):
1. "The goal of this review is to describe which tools or domains are able to critically evaluate the quality of a CP along different domains, in order to feed up further research devoted to design and implement an HTA framework for CP." Why did the authors consider conducting an overview of systematic reviews to achieve this goal? A justification for this is missing.
2. What is the research question of this overview? It should be explicitly described in the manuscript. Consider this reference: Hunt, H., Pollock, A., Campbell, P. et al. An introduction to overviews of reviews: planning a relevant research question and objective for an overview. Syst Rev 7, 39 (2018). https://doi.org/10.1186/s13643-018-0695-8
3. The term "Pathway*[tiab]" could be removed from the search string because there is "path* [tiab]".
4. Consider "in-vitro and animal studies and conference abstracts were excluded". a) Only secondary studies were not included? This criterion would be unnecessary. b) Are there systematic reviews published as conference abstracts?
5. Why was "publication date in the last 5 years" used as selection criterion? When was the search conducted? Moreover, why were 6 studies published before 2015 included? If snowballing was used to include reviews published out of this period, so this selection criterion should not be applied.
6. I could not understand anything in section "Statistical analysis". Text should be revised.
7. The explanation on the "Overall rating quality" below table 3 should be in the text, not as legend.
8. What is new in this paper? I mean, what is different from the published literature? The results and discussions presented seems to be a consensus.
9. There are some huge paragraphs in the manuscript (e.g., in introduction and discussion sections). Please split them.

Specific comments:
- There are some formatting mistakes (words/sentences together). This did not avoid understanding the paper, but should be corrected.
- lines 20-21: "...teammultidisciplinary involvement/team..." -> ??? team 2 times
- Sections and subsections should be numbered.
- lines 62-65: not closed parenthesis
- particolar -> particUlar
- Don not use abbreviations (e.g., can't -> cannot).
- Abbreviation HTA is defined only in discussion. All abbreviations should be defined at the first time they are used.
- "very literature" -> much
...
I strongly recommend an English review looking for typos and grammar errors.

Author Response

Please see the table on the file

Round 2

Reviewer 1 Report

Thank you for reworking this important manuscript. I appreciate the authors’ clarification of purpose and explanation of how a gold standard for clinical pathway construction and evaluation could be conceived. I especially like the figure that illustrates the conceptual framework.

Congratulations on an excellent contribution to this literature.

Author Response

Dear Reviewer, 

We are deeply grateful for the time devoted to our work, and for your suggestions 

Kind Regards

All Authors

Reviewer 2 Report

The authors answered all of my questions and addressed my concerns.

I have 2 additional minor comments.

  1. Title change suggestion -> Towards a New System for the Assessment of the Quality in Care Pathways: An Overview of Systematic Reviews
  2. Line 145 is orphan ("Prospero Protocol approved (CRD42020210486)". The manuscript is not a checklist. I suggest that some text be added to the sentence to make more sense. For example, "We recorded the protocol for this review in the PROSPERO, which was approved.... etc"

Author Response

Dear Reviever, 

We have changed the title as suggested, and rewrote of the sentence of the PROPERO protocol.

We are deeply grateful 

Thanks

This manuscript is a resubmission of an earlier submission. The following is a list of the peer review reports and author responses from that submission.